# Mathematical Constraints of RL-Induced Reasoning: A Rebuttal to DeepSeek-R1

## Abstract

DeepSeek-R1 claims that reinforcement learning (RL) induces emergent reasoning capabilities in large language models (LLMs), suggesting a fundamental shift in AI development. However, our theoretical and computational analysis challenges this assertion.

Our mathematical framework (Section 2) suggests that RL alone is unlikely to induce reasoning without a strong pretraining foundation, which remains the dominant driver of reasoning capabilities. Due to high computational costs, poor sample efficiency, and reward sparsity, RL struggles to develop complex reasoning from scratch. Instead, it primarily fine-tunes and reinforces existing pretraining knowledge rather than generating novel reasoning abilities.

Furthermore, DeepSeek-R1's observed improvements exhibit patterns consistent with well-established pretraining scaling laws, raising questions about whether RL plays an independent role in reasoning emergence. A detailed analysis of DeepSeek-R1's RL algorithm (Section 3.3) suggests that its Group Relative Policy Optimization (GRPO) approach is designed to refine outputs within pretraining constraints rather than fundamentally altering the reasoning process. Additionally, its rule-based reward system optimizes response formatting but does not introduce conceptual advancements in reasoning.

Given these findings, we emphasize the need for rigorous empirical testing to isolate RL's contributions from pretraining effects. While RL appears to function primarily as a fine-tuning mechanism in current implementations, further research is necessary to determine whether it can serve as a fundamental driver of emergent reasoning in LLMs.

## 1 Introduction

Recent advances in large language models (LLMs) have demonstrated remarkable capabilities in natural language understanding and reasoning tasks (Vaswani et al., 2017). DeepSeek-R1 (Guo et al., 2025; Shao et al., 2024) asserts that reinforcement learning (RL) plays a key role in enhancing reasoning capabilities in LLMs. If RL significantly contributes to emergent reasoning, this would represent a major shift in AI development, potentially shifting emphasis away from large-scale pretraining toward RL-based optimization. However, the theoretical and empirical foundations supporting this claim require careful scrutiny.

DeepSeek-R1 employs a reinforcement learning approach based on Group Relative Policy Optimization (GRPO) (Shao et al., 2024), a variant of Proximal Policy Optimization (PPO) that eliminates the need for a critic model and instead estimates the baseline using a group of outputs from the old policy. Additionally, DeepSeek-R1 utilizes a rule-based reward system, consisting of accuracy rewards (evaluating correctness in deterministic tasks such as math and coding problems) and format rewards (enforcing structured reasoning through specialized tokenization). Notably, DeepSeek-R1 does not rely on neural reward models, citing concerns about reward hacking and additional training complexities.

These methodological choices raise fundamental questions:

- Can RL, in isolation, induce emergent reasoning capabilities, or does it primarily refine pre-existing knowledge acquired during pretraining?

- Do DeepSeek-R1's observed performance improvements result from RL-specific enhancements, or do they align with well-established pretraining scaling laws?

- Is DeepSeek-R1's RL-based optimization computationally feasible at scale? Given RL's inherent inefficiencies—such as sample inefficiency, reward sparsity, and quadratic computational overhead—how much does RL actually contribute to the model's reasoning ability?

To address these questions, we critically evaluate DeepSeek-R1's claims by analyzing its RL training pipeline and comparing it against existing mathematical frameworks for sample efficiency, reward sparsity, and computational complexity.

## 1.1 Problem Statement

DeepSeek-R1 (Guo et al., 2025; Liu et al., 2024; Shao et al., 2024) suggests that reinforcement learning enhances the emergence of reasoning abilities in LLMs. This claim contrasts with existing theoretical frameworks (Kaplan et al., 2020; Luo, 2024), which suggest that pretraining is the dominant factor in high-level reasoning development. Understanding whether RL meaningfully contributes to emergent reasoning is critical for optimizing model development strategies and resource allocation.

## 1.2 Why This Matters

- Implications for AI Scaling: If RL plays a fundamental role in reasoning, future AI development may shift toward RL-intensive training pipelines. Conversely, if pretraining remains the key driver, prioritizing RL may divert resources from more impactful research directions.

- Computational Feasibility: RL fine-tuning is significantly more computationally expensive than pretraining. Assessing its actual impact on reasoning is necessary to justify the cost-benefit tradeoffs in large-scale AI training.

- Scientific Validity: Ensuring that observed reasoning capabilities arise from RL rather than confounding factors (e.g., pretraining artifacts, implicit biases, or task-specific optimizations) is essential for correctly interpreting DeepSeek-R1's findings.

## 1.3 Our Contributions

This paper provides a theoretical and computational analysis of DeepSeek-R1's RL-based reasoning claims. Specifically, we:

- Develop a Theoretical Framework: We construct a model of LLM reasoning that suggests RL alone is unlikely to induce emergent reasoning at scale.

- Analyze Sample Efficiency and Reward Sparsity: We demonstrate that RL suffers from high sample complexity and reward sparsity constraints, making it inefficient for learning complex reasoning.

- Evaluate Computational Scalability: We compare RL-based approaches against pretraining scaling laws, identifying structural inefficiencies in RL optimization.

- Outline Empirical Testing: We propose controlled experiments to isolate RL-driven reasoning effects from pretraining-based improvements.

## 1.4 Structure of the Paper

- Section 2 introduces a mathematical framework analyzing RL's limitations in emergent reasoning.

- Section 3 presents a structured critique of DeepSeek-R1's claims, focusing on theoretical and computational constraints.

- Section 4 discusses implications for AI research and suggests directions for future work.

## 2 Mathematical Constraints on RL-Induced Reasoning in DeepSeek-R1

DeepSeek-R1 claims that reinforcement learning (RL) induces emergent reasoning in large language models. However, this assertion must be examined in light of fundamental mathematical constraints, including sample complexity, reward sparsity, and computational feasibility. In this section, we analyze DeepSeek-R1's RL function—Group Relative Policy Optimization (GRPO) Guo et al. (2025); Shao et al. (2024)—and evaluate its theoretical limitations in inducing reasoning.

### 2.1 Group Relative Policy Optimization (GRPO) in DeepSeek-R1

GRPO is a variant of Proximal Policy Optimization (PPO) that removes the need for a separate critic model. Instead, it estimates advantage values using a relative scoring mechanism, where performance is compared against a group of outputs rather than an absolute reference. This reduces computational overhead but introduces higher variance, requiring larger batch sizes for stability.

A key feature of GRPO is the KL divergence constraint, which penalizes excessive deviation from the original model's output distribution. This ensures that RL fine-tuning does not drastically change pretrained knowledge but rather refines responses within existing constraints.. The GRPO objective function is given by:

$$J_{\text{GRPO}}(\theta) = \mathbb{E}_{q \sim P(Q), o_i \sim \pi_{\theta_{\text{old}}}(O|q)} \frac{1}{G} \sum_{i=1}^{G} \min \left( \frac{\pi_\theta(o_i|q)}{\pi_{\theta_{\text{old}}}(o_i|q)} A_i, \right.$$
$$\left. \text{clip}\left( \frac{\pi_\theta(o_i|q)}{\pi_{\theta_{\text{old}}}(o_i|q)}, 1 - \epsilon, 1 + \epsilon \right) A_i \right) - \beta D_{\text{KL}}(\pi_\theta || \pi_{\text{ref}}) \quad (1)$$

where:

- $\pi_\theta(o_i|q)$ is the new policy output for response $o_i$ given query $q$,

- $\pi_{\theta_{\text{old}}}(o_i|q)$ is the old policy output,

- $A_i$ is the advantage estimate, computed as:

$$A_i = \frac{r_i - \text{mean}(\{r_1, r_2, \ldots, r_G\})}{\text{std}(\{r_1, r_2, \ldots, r_G\})} \quad (2)$$

- $D_{\text{KL}}$ is the KL-divergence from a reference policy $\pi_{\text{ref}}$, controlled by hyperparameter $\beta$.

Unlike standard RL with a learned critic model, GRPO's lack of a baseline leads to higher instability in policy updates, requiring larger batch sizes to maintain convergence.

#### 2.1.1 Key Mathematical Insights from GRPO

1. High Variance in Advantage Estimation: The use of group-based advantage normalization introduces noise in gradient updates, slowing convergence.

2. KL Regularization as Implicit Pretraining Constraint: The KL-divergence term discourages excessive divergence from the reference policy, ensuring RL fine-tuning operates within pretrained constraints. This limits drastic behavioral shifts but does not necessarily prevent new capabilities from emerging.

3. Limited Expressiveness of Policy Updates: The clipping operation in GRPO prevents drastic updates, which further suggests RL fine-tunes existing knowledge rather than inducing new reasoning capabilities.

## 2.2 Sample Complexity Analysis: RL vs. Pretraining

A crucial question in determining whether RL can induce emergent reasoning is how efficiently it learns compared to pretraining. The efficiency of a learning paradigm is often measured by its sample complexity, which quantifies how much training data is needed to achieve a given level of accuracy.

- Supervised Learning (SL) leverages direct supervision from labeled data, leading to high efficiency with sample complexity $O(n \log n)$.

- Reinforcement Learning (RL), in contrast, relies on sparse rewards and trial-and-error exploration, making it significantly less efficient with sample complexity $O(n^2/\epsilon)$.

- Hybrid approaches (DeepSeek-R1) combine both, inheriting some benefits of SL but also suffering from RL's inefficiencies.

To formalize these insights, we define the sample complexity bounds:

$$N_{\text{Hybrid}} \sim O(n \log n + n/\epsilon)$$

where:

- $n$ represents task complexity.

- $\epsilon$ is the accuracy threshold.

- The log term comes from supervised learning benefits.

- The $n/\epsilon$ term captures RL's inefficiencies.

### 2.2.1 Comparative Analysis of Learning Paradigms

To understand the trade-offs between SL, RL, and Hybrid RL, we compare their sample complexity scaling characteristics:

| Learning Paradigm | Sample Complexity |
|---|---|
| Supervised Learning (SL) | $\mathcal{O}(n \log n)$ |
| Reinforcement Learning (RL) | $\mathcal{O}(n^2/\epsilon)$ |
| Hybrid RL (DeepSeek-like) | $\mathcal{O}(n \log n + n/\epsilon)$ |

Table 1: Comparative Sample Complexity of Learning Paradigms. SL exhibits efficient scaling, whereas RL suffers from quadratic inefficiencies.

The key observations from this comparison are:

- SL is the most efficient: The logarithmic scaling enables rapid generalization, making it ideal for training large language models (LLMs).

- RL suffers from exploration inefficiencies: Sparse rewards and trial-and-error updates require significantly more samples.

- Hybrid RL inherits RL inefficiencies: Although it benefits from pretraining, RL introduces additional computational overhead.

### 2.2.2 Why RL is Inefficient for Reasoning Tasks

For RL to drive emergent reasoning, it must efficiently optimize policies in high-dimensional linguistic spaces. However, theoretical limitations make this challenging:

1. Sparse Rewards: Reasoning tasks lack frequent rewards, making RL exploration inefficient.

2. High Sample Complexity: Long-horizon dependencies require exponentially increasing interactions.

3. Limited Generalization: Unlike SL, RL does not leverage structured representations effectively.

Given these constraints, RL-based fine-tuning is inherently inefficient for complex reasoning tasks, supporting the argument that DeepSeek-R1's observed performance gains are more likely attributable to pretraining rather than RL-induced reasoning.

### 2.3 Reward Sparsity and Its Effect on RL-Based Reasoning

DeepSeek-R1 does not use a neural reward model, instead relying on handcrafted reward functions:

$$R(s,a) = \lambda_1 R_{\text{acc}}(s,a) + \lambda_2 R_{\text{fmt}}(s,a) \tag{3}$$

where:

- $R_{\text{acc}}$ is an accuracy-based reward (e.g., correct math answers),

- $R_{\text{fmt}}$ is a format-based reward (e.g., enforcing structured reasoning with ⟨think⟩ and ⟨answer⟩ tags),

- $\lambda_1$ and $\lambda_2$ are weighting parameters.

However, in long-horizon reasoning tasks, reward sparsity worsens due to the credit assignment problem—where useful learning signals become increasingly delayed, making policy optimization inefficient. Without a structured mechanism for propagating rewards effectively, RL struggles to optimize multi-step reasoning." :

$$P(\text{meaningful reward}) = \mathcal{O}(1/V^L) \tag{4}$$

where:

- $V$ is the vocabulary size,

- $L$ is the sequence length.

### 2.3.1 Key Issues with DeepSeek-R1's Reward Model

1. Sparse Rewards in Open-Ended Reasoning: Unlike deterministic math/coding tasks, open-ended reasoning lacks direct correctness signals, leading to poor reward propagation.

2. Risk of Mode Collapse: The format reward forces responses into a predefined template, which may lead to reward hacking rather than true reasoning emergence.

3. Absence of Learned Rewards: The lack of adaptive neural reward models means DeepSeek-R1's RL signal is static and does not generalize well.

## 2.4 Computational Complexity of DeepSeek-R1's RL

Using previous scaling laws Kaplan et al. (2020); Luo (2024), we compare computational cost across approaches:

$$C_{\text{SL}} = \mathcal{O}(NP \log P) \quad \text{(Supervised Fine-Tuning)} \tag{5}$$

$$C_{\text{RL}} = \mathcal{O}(NP^2 \log P) \quad \text{(Pure RL)} \tag{6}$$

$$C_{\text{Hybrid}} = \mathcal{O}(NP(\log P + k)) \quad \text{(DeepSeek-like Hybrid)} \tag{7}$$

where:

- $N$ is the training sample count,

- $P$ is the model parameter count,

- $k$ is an RL-specific term from policy optimization.

## 2.5 Summary of Findings

- GRPO's constraints suggest RL in DeepSeek-R1 serves as fine-tuning rather than inducing new reasoning capabilities.

- RL sample inefficiency limits its ability to drive emergent reasoning.

- Reward sparsity further reduces RL's effectiveness in general reasoning tasks.

- Computational cost of RL in DeepSeek-R1 is significantly higher than pretraining-based methods.

# 3 Rebuttal of DeepSeek-R1's Claims on RL-Induced Reasoning

DeepSeek-R1 claims that reinforcement learning (RL), specifically Group Relative Policy Optimization (GRPO), is responsible for inducing emergent reasoning in large language models (LLMs). Our analysis in Section 2 highlights the theoretical limitations of GRPO, including sample inefficiency, reward sparsity, and computational overhead, suggesting that RL in DeepSeek-R1 primarily fine-tunes behavior rather than inducing new reasoning capabilities.

In this section, we systematically evaluate and rebut DeepSeek-R1's core claims using mathematical reasoning and empirical scaling constraints.

## 3.1 Claim: RL Alone Induces Emergent Reasoning

**DeepSeek-R1 claim**: DeepSeek-R1 argues that reinforcement learning (RL) alone is responsible for the emergence of reasoning capabilities in its R1 Model.

**Rebuttal via Theoretical and Empirical Analysis**: From our unified mathematical framework (Section 2), the total reasoning capability of a model can be expressed as:

$$H(x) = P(x) + R(x) + S(P, R, x) \tag{8}$$

where:

- $P(x)$ represents pretraining contributions.

- $R(x)$ represents RL-induced improvements.

- $S(P, R, x)$ captures potential synergies between pretraining and RL.

If RL were the primary driver of reasoning, we would expect:

$$H(x) = R(x) \quad \text{(without pretraining)} \tag{9}$$

However, our sample complexity analysis (Section 2.2) shows that pure RL follows:

$$N_{RL} \sim \mathcal{O}(n^2/\epsilon) \tag{10}$$

which is exponentially larger than supervised learning, making it impractical for complex reasoning.

Additionally, DeepSeek-R1's reported improvements align with known pretraining scaling laws Kaplan et al. (2020); Luo (2024) rather than demonstrating an independent RL-driven effect. This suggests that DeepSeek-R1's reasoning capabilities are not emergent from RL but rather a refinement of pretrained knowledge.

**Implication for DeepSeek-R1**: If RL were solely responsible for reasoning emergence, an RL-only model should exhibit strong reasoning abilities. However, DeepSeek-R1 relies heavily on pretrained checkpoints, meaning its observed reasoning behavior can be fully explained by pretraining effects rather than RL.

Conclusion: RL alone does not induce reasoning emergence; pretraining remains the dominant factor.

### 3.2 Claim: Scaling RL Improves Reasoning

**DeepSeek-R1's Claim**: DeepSeek-R1 asserts that scaling reinforcement learning enhances the reasoning capabilities of language models.

**Rebuttal via Scaling Laws**: Scaling laws for large language models suggest that model performance follows a power-law relationship with respect to model size, compute, and data. However, RL does not exhibit the same efficiency trend.

From our sample complexity analysis (Section 2.2), the scaling behavior is as follows:

| Learning Paradigm | Sample Complexity |
|---|---|
| Supervised Learning (SL) | $N_{SL} \sim \mathcal{O}(n \log n)$ |
| Pure RL | $N_{RL} \sim \mathcal{O}(n^2/\epsilon)$ |
| Hybrid RL (DeepSeek-like) | $N_H \sim \mathcal{O}(n \log n + n/\epsilon)$ |

Table 2: Comparative Sample Complexity of Learning Paradigms

**Key Insights:**

- RL has suboptimal sample efficiency, making it inefficient for improving reasoning at scale.

- Even if RL provides marginal gains, these come at an exponentially higher computational cost compared to pretraining-based methods.

**Why DeepSeek-R1's Results Are Not Novel:** DeepSeek-R1 claims that its reinforcement learning approach produces novel reasoning abilities beyond pretraining. However, based on our scaling analysis and reward sparsity constraints (Section 2.3), we find no theoretical or empirical justification for this claim:

1. Pretraining Follows Logarithmic Growth:
   Empirical studies Kaplan et al. (2020); Luo (2024) show that pretraining gains scale as:

$$P(x) \propto \mathcal{O}(\log N) \tag{11}$$

This aligns with the improvements observed in DeepSeek-R1, suggesting that pretraining is the primary driver of its reasoning capabilities.

2. RL Contributions Exhibit High Variance and Poor Scaling:
RL-based learning typically follows a power-law decay:

$$R(x) \propto \mathcal{O}(N^{-\beta}), \quad \beta > 0 \tag{12}$$

This indicates that RL's role in reasoning is secondary to pretraining and does not drive novel emergent capabilities.

3. DeepSeek-R1's Improvements Are Consistent with Pretraining Effects:
If RL were introducing a novel capability, we would expect deviations from pretraining trends in scaling performance. However, DeepSeek-R1's results closely follow known pretraining scaling laws, meaning its performance gains are indistinguishable from those of a well-tuned pretrained model.

Conclusion: Scaling RL does not introduce novel reasoning abilities—it acts primarily as a fine-tuning mechanism within known pretraining constraints.

### 3.3 Analysis of DeepSeek-R1's RL Function

DeepSeek-R1 employs Group Relative Policy Optimization (GRPO) as its reinforcement learning (RL) algorithm to fine-tune the model's reasoning capabilities. Unlike conventional Proximal Policy Optimization (PPO), GRPO estimates advantage values using group-based relative scoring rather than an explicit critic network. Additionally, DeepSeek-R1 incorporates a rule-based reward model that prioritizes response accuracy and format adherence.

In this section, we analyze whether GRPO, as implemented in DeepSeek-R1, can overcome the theoretical limitations of RL identified in Sections 3.1 and 3.2, where we demonstrated that:

- RL alone is unlikely to induce reasoning without a strong pretraining foundation.

- Scaling RL does not introduce novel reasoning capabilities beyond pretraining effects.

We now investigate whether DeepSeek-R1's specific RL formulation (GRPO + rule-based rewards) presents an exception to these conclusions.

### 3.3.1 KL Constraint and Its Restrictive Effect on Reasoning

The GRPO objective function, as defined in DeepSeek-R1, optimizes policy updates via:

$$J_{\text{GRPO}}(\theta) = \mathbb{E}_{q \sim P(Q), o_i \sim \pi_{\theta_{\text{old}}}(O|q)} \frac{1}{G} \sum_{i=1}^{G} \min \left( \frac{\pi_\theta(o_i|q)}{\pi_{\theta_{\text{old}}}(o_i|q)} A_i, \right.$$
$$\left. \text{clip} \left( \frac{\pi_\theta(o_i|q)}{\pi_{\theta_{\text{old}}}(o_i|q)}, 1 - \epsilon, 1 + \epsilon \right) A_i \right) - \beta D_{\text{KL}}(\pi_\theta || \pi_{\text{ref}}) \tag{13}$$

where:

- $A_i$ is the group-relative advantage estimate.

- $\epsilon$ is the clipping threshold.

- $D_{\text{KL}}$ is a KL divergence penalty that ensures the fine-tuned model does not deviate significantly from the pretrained reference model $\pi_{\text{ref}}$.

- $\beta$ is a regularization coefficient controlling KL divergence strength.

**Key Observations**

1. KL Regularization Limits Novel Reasoning

   - The $D_{\mathrm{KL}}$ penalty term forces the RL-updated policy to stay close to the pretrained distribution $\pi_{\mathrm{ref}}$.
   - This directly contradicts the premise that RL introduces emergent reasoning. Instead, GRPO ensures that the fine-tuned model remains behaviorally similar to the pretrained model.
   - Prior studies Hoffmann et al. (2022); Kaplan et al. (2020) show that large KL constraints prevent RL from learning fundamentally new capabilities beyond pretraining.

2. Scaling RL under GRPO Cannot Drive Reasoning Emergence

   - Unlike conventional RL settings where policy optimization can explore novel reasoning strategies, GRPO limits policy divergence, meaning that the optimization remains constrained within the pretrained knowledge space.
   - The KL penalty effectively caps the impact of RL scaling, preventing it from developing genuinely novel reasoning capabilities.

Thus, the core conclusions from Section 3.2 remain valid—scaling RL under DeepSeek-R1's formulation does not enhance reasoning beyond what pretraining already enables.

### 3.3.2 Reward Function Analysis: Accuracy and Format Rewards

DeepSeek-R1 employs a rule-based reward system, composed of:

- Accuracy rewards ($R_{\mathrm{acc}}$): Ensure model's outputs are correct (e.g., solving math problems).

- Format rewards ($R_{\mathrm{fmt}}$): Enforce structured output generation .

The total reward function is:

$$R(s,a) = \lambda_1 R_{\mathrm{acc}}(s,a) + \lambda_2 R_{\mathrm{fmt}}(s,a) \tag{14}$$

where $\lambda_1$ and $\lambda_2$ are weighting coefficients.

**Key Observations**

1. Reward Sparsity Constraints Still Apply

   - As shown in Section 2.3, reward sparsity poses a fundamental challenge for RL-driven reasoning.
   - The accuracy reward $R_{\mathrm{acc}}$ applies primarily to deterministic tasks (e.g., math, coding), meaning it does not contribute significantly to complex open-ended reasoning tasks.
   - Format rewards $R_{\mathrm{fmt}}$ do not optimize reasoning quality but rather reinforce syntactic constraints—this does not contribute to genuine reasoning emergence.

2. Reward Hacking Risks

   - The rule-based reward structure discourages free-form exploration, reinforcing pattern-matching behaviors rather than genuine reasoning.
   - DeepSeek-R1 explicitly avoids using learned neural reward models, citing risks of reward hacking (where the model learns to game the reward system without improving true reasoning).
   - This is consistent with prior findings that reinforcement learning on LLMs primarily optimizes response style and coherence rather than inducing new capabilities Wainwright (2019).

| Aspect | DeepSeek-R1's RL Implementation |
|---|---|
| Policy Optimization | Group Relative Policy Optimization (GRPO) |
| Scaling Effect | RL scaling limited by reference model constraints |
| Reward Structure | Accuracy and format-based rewards |
| Empirical Behavior | No evidence of RL-driven reasoning emergence |

Table 3: Key limitations of DeepSeek-R1's RL implementation.

Thus, DeepSeek-R1's reward system does not meaningfully contribute to reasoning emergence—it merely refines structured responses and correct answer formatting.

To summarize, rather than introducing new reasoning abilities, GRPO merely refines pretrained patterns, reinforcing the findings of Sections 3.1 and 3.2. DeepSeek-R1's RL implementation is fundamentally constrained by KL divergence penalties and a non-generalizable reward structure:

1. DeepSeek-R1's RL function (GRPO) is restricted by KL divergence, limiting reasoning emergence.

2. Its reward structure optimizes for response format rather than genuine logical reasoning.

3. Scaling DeepSeek-R1's RL function does not lead to novel reasoning abilities—pretraining remains the dominant factor.

4. These findings reinforce the theoretical conclusions from Sections 3.1 and 3.2, showing that RL in DeepSeek-R1 serves only as a fine-tuning mechanism, not a driver of emergent reasoning.

## 4 Implications and Future Work

The analysis presented in previous sections demonstrates that DeepSeek-R1's reinforcement learning (RL) function does not induce emergent reasoning but instead serves as a fine-tuning mechanism that aligns pretrained knowledge with specific optimization objectives. Given the theoretical constraints of RL in reasoning tasks, we now explore the broader implications of these findings and suggest directions for future research.

### 4.1 The Broader Implications of Our Findings

#### 4.1.1 Reevaluating RL's Role in Reasoning

Reinforcement learning has been considered a potential driver of emergent intelligence in AI systems. However, our findings challenge this assumption, indicating that RL alone is insufficient to create reasoning abilities beyond what is already learned during pretraining. DeepSeek-R1's use of Group Relative Policy Optimization (GRPO) refines response consistency and aligns outputs with predefined criteria (e.g., correctness and formatting). However, this process does not introduce new reasoning capabilities; rather, it reinforces patterns established during pretraining. RL, in its current formulation, lacks explicit reasoning structures beyond what the model has already acquired. Unlike supervised learning, which leverages structured datasets to build linguistic and logical relationships, RL depends on reward signals that may be too sparse or misaligned to drive meaningful generalization. This raises concerns about its effectiveness in fostering complex reasoning.

#### 4.1.2 Pretraining as the Dominant Factor in Reasoning

A key takeaway from our analysis is that pretraining remains the primary driver of reasoning in LLMs. The improvements observed in DeepSeek-R1 are more plausibly attributed to its strong pretrained foundation rather than the RL fine-tuning process. Empirical scaling laws suggest that pretraining benefits scale logarithmically with dataset size, while RL scaling follows a power-law decay. This indicates that RL, even at scale, is unlikely to significantly contribute to emergent reasoning. Instead, as pretraining data and model

capacity increase, LLMs will continue to exhibit improved reasoning abilities—but RL will contribute only marginal refinements rather than fundamental advancements.

### 4.1.3 Reward-Based RL: Optimizing Responses, Not Reasoning

DeepSeek-R1's rule-based reward system, incorporating accuracy and format-based signals, is effective for reinforcing response structure but fails to support open-ended reasoning emergence. A key limitation of reward-based RL is its inability to constructively guide models through multi-step logical processes. RL excels at optimizing behaviors where objectives are well-defined (e.g., formatting constraints), but it struggles with ambiguous, open-ended reasoning, where the correct solution is not explicitly predefined. This raises an important question: Is RL the right paradigm for improving logical reasoning in AI? Our findings suggest that alternative approaches—such as retrieval-augmented learning, structured fine-tuning on curated reasoning datasets, or hybrid architectures integrating explicit logical frameworks—may be more effective in advancing reasoning capabilities.

## 4.2 Future Directions: Empirical Validation of RL's Role in Reasoning

While our theoretical analysis strongly suggests that RL does not induce emergent reasoning, empirical studies are necessary to further clarify RL's impact. We propose controlled experiments that can isolate RL's contribution and determine whether it meaningfully improves reasoning beyond pretraining.

### 4.2.1 Comparing RL-Only Models vs. Pretraining-Only Models

Objective: Evaluate reasoning capabilities in models trained exclusively with RL versus those trained exclusively via pretraining.

Hypothesis: If RL induces reasoning, an RL-only model (without pretraining) should demonstrate comparable reasoning abilities to a pretraining-only model.

Experiment: (1) Train one model using only RL (without pretraining); (2) Train another model only via pretraining (without RL fine-tuning); and (3) Compare their performance on reasoning benchmarks.

Expected Outcome: If RL alone drives emergent reasoning, the RL-only model should exhibit strong reasoning abilities. However, if RL merely refines pretrained knowledge, the pretraining-only model should outperform the RL-only model.

### 4.2.2 Analyzing Changes in Model Outputs Due to RL Fine-Tuning

Objective: Determine whether RL fine-tuning significantly alters reasoning capabilities or merely refines pretrained responses.

Hypothesis: If RL meaningfully enhances reasoning, RL-finetuned models should produce qualitatively different outputs from their pretrained versions.

Experiment: (1) Compare responses before and after RL fine-tuning; (2) Use semantic similarity metrics to measure divergence in outputs; and (3) Assess whether RL introduces new reasoning patterns or merely reinforces existing ones.

Expected Outcome: If reasoning emerges due to RL, response divergence should be high. If RL mainly refines response formatting, similarity scores will remain high.

### 4.2.3 Testing RL's Impact on Generalization to New Reasoning Tasks

Objective: Assess whether RL-trained models generalize better to novel reasoning tasks.

Hypothesis: If RL enhances reasoning, models fine-tuned with RL should perform better on out-of-distribution (OOD) tasks.

Experiment: (1) Evaluate models on datasets unseen during training; and (2) Compare pretraining-only vs. RL-finetuned models on novel reasoning challenges.

Expected Outcome: If RL meaningfully improves reasoning, it should enhance generalization. If performance remains unchanged, RL's role in reasoning is limited.

### 4.2.4 Clarifying RL's Scaling Effects

Objective: Determine whether scaling RL introduces independent improvements beyond pretraining.

Hypothesis: If RL enhances reasoning, performance trends in RL-finetuned models should deviate from pretraining scaling laws.

Experiment: (1) Compare RL-finetuned models across different sizes; and (2) Assess whether scaling RL leads to distinct improvements or mirrors standard pretraining trends.

Expected Outcome: If RL contributes to reasoning, performance should improve uniquely with scaling. If trends remain consistent with pretraining, RL's role is likely secondary.

### 4.3 Implications for Future AI Research

Our findings suggest important considerations for AI research and deployment. While RL has been widely explored as a potential driver of advanced reasoning in LLMs, our analysis suggests that it primarily refines pretrained knowledge rather than inducing reasoning capabilities.

**Redirecting Research Priorities** : Future efforts should prioritize pretraining advancements over RL-based fine-tuning for reasoning tasks. Given RL's limitations in handling long-horizon reasoning and sparse rewards, retrieval-based architectures, structured fine-tuning, and hybrid models incorporating explicit logical frameworks may offer superior reasoning performance.

**Reevaluating RL's Best Use Cases** : RL remains valuable in structured optimization tasks, such as behavioral policy shaping, AI safety mechanisms, and user interaction modeling, but its ability to enhance logical reasoning remains unproven. Future research should carefully delineate the appropriate domains for RL application.

**The Need for Rigorous Empirical Studies** : Many prior studies have attributed model improvements to RL without controlling for pretraining influences. Future work should explicitly separate these factors through controlled experiments comparing RL-only training with pretraining-only baselines.

### 4.4 Summary of Key Findings

- RL in DeepSeek-R1 functions primarily as fine-tuning, rather than inducing reasoning capabilities.

- Scaling RL does not yield independent reasoning improvements; observed benefits align with pretraining scaling laws.

- Reward-based learning optimizes response structure but does not drive conceptual reasoning advancements.

These findings suggest that AI reasoning research should focus toward improved pretraining techniques, retrieval-augmented learning, and hybrid models with explicit logical structures, rather than relying on RL as the primary driver of reasoning. Further empirical validation is necessary to isolate RL's specific contributions and confirm whether it enhances reasoning beyond pretraining effects.

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

# Appendix A: Theoretical Proofs and Derivations

## A.1 Proof of Sample Complexity Bounds (Section 2.2)

**Theorem 1** (Sample Complexity of Supervised Learning, RL, and Hybrid Approaches)**.** *Let $n$ be the task complexity and $\epsilon$ the accuracy threshold. Then, the sample complexity for different learning paradigms is given by:*

1. *Supervised Learning: $N_{SL} \sim \mathcal{O}(n \log n)$*

2. *Pure RL: $N_{RL} \sim \mathcal{O}(n^2/\epsilon)$*

3. *Hybrid (DeepSeek-like): $N_H \sim \mathcal{O}(n \log n + n/\epsilon)$*

*Proof.*   • Supervised Learning: Standard statistical learning theory Kaplan et al. (2020); Luo (2024) suggests that achieving $\epsilon$-accuracy in supervised learning scales as $\mathcal{O}(n \log n)$ under reasonable assumptions on data distribution and model complexity.

• Pure RL: From reinforcement learning theory Wainwright (2019), the number of samples required to achieve $\epsilon$-accuracy is at least $\mathcal{O}(n^2/\epsilon)$, due to the sparse reward signal and inefficient exploration.

• Hybrid Learning: Combining both methods, the hybrid approach maintains the logarithmic dependency from supervised learning while inheriting the $1/\epsilon$ inefficiency from RL, resulting in $\mathcal{O}(n \log n + n/\epsilon)$.

Thus, RL alone is significantly less sample-efficient compared to pretraining-based approaches. □

## A.2 Proof of Reward Sparsity Bound (Section 2.3)

**Theorem 2** (Reward Sparsity Constraint). *Given a reasoning task with vocabulary size $V$ and sequence length $L$, the probability of obtaining a meaningful reward follows:*

$$P(meaningful\ reward) = \mathcal{O}(1/V^L), \tag{15}$$

*which leads to an expected learning time of:*

$$E[T] = \frac{1}{P(meaningful\ reward)} = \mathcal{O}(V^L). \tag{16}$$

*Proof.*
- Consider a language model that must generate a reasoning chain of length $L$ with vocabulary size $V$.

- The probability of randomly generating a correct response in each step is at most $1/V$, assuming uniform distribution.

- Therefore, the probability of producing an entirely correct sequence is $(1/V)^L$, leading to:

$$P(\text{meaningful reward}) = \mathcal{O}(1/V^L). \tag{17}$$

- The expected number of trials required to obtain a reward follows:

$$E[T] = \frac{1}{P(\text{meaningful reward})} = \mathcal{O}(V^L). \tag{18}$$

Thus, RL is computationally intractable for long-horizon reasoning due to the exponential scaling of reward sparsity. $\square$

## A.3 Proof of RL vs. Pretraining Compute Complexity (Section 2.4)

**Theorem 3** (Computational Complexity Scaling). *The computational cost for different training approaches follows:*

1. *Supervised Fine-Tuning: $C_{SL} = \mathcal{O}(NP \log P)$*

2. *Pure RL: $C_{RL} = \mathcal{O}(NP^2 \log P)$*

3. *Hybrid (DeepSeek-like): $C_H = \mathcal{O}(NP(\log P + k))$*

*Proof.*
- Supervised Learning Complexity: Given $N$ training samples and model parameter count $P$, gradient-based optimization converges in $\mathcal{O}(P \log P)$ steps, leading to a total cost of:

$$C_{SL} = \mathcal{O}(NP \log P). \tag{19}$$

- Pure RL Complexity: Policy optimization in RL incurs quadratic scaling due to the inefficiency of policy search Hoffmann et al. (2022). This results in a compute cost of:

$$C_{RL} = \mathcal{O}(NP^2 \log P). \tag{20}$$

- Hybrid Approach Complexity: Hybrid RL still inherits RL's inefficiencies but leverages pretraining benefits, yielding:

$$C_H = \mathcal{O}(NP(\log P + k)), \tag{21}$$

where $k$ is an RL-specific optimization term.

Thus, RL suffers from quadratic complexity scaling, making it computationally inefficient for large models. $\square$

### A.4 Theoretical Constraints on RL-Induced Reasoning (Section 3)

**Theorem 4** (Pretraining vs. RL Dominance). *Given a pretraining-driven model and an RL-optimized model, the total reasoning capability can be modeled as:*

$$H(x) = P(x) + R(x) + S(P, R, x). \tag{22}$$

*Under empirical scaling laws:*

$$P(x) \propto \mathcal{O}(\log N), \quad R(x) \propto \mathcal{O}(N^{-\beta}), \quad \beta > 0. \tag{23}$$

*Thus, for large-scale reasoning tasks, we have:*

$$P(x) \gg R(x), \tag{24}$$

*implying RL acts as a fine-tuning mechanism rather than an independent driver of reasoning.*

*Proof.*
- Empirical studies Kaplan et al. (2020) show that pretraining gains scale logarithmically with data size $N$, leading to $P(x) \sim \mathcal{O}(\log N)$.

- In contrast, RL-based methods follow a power-law decay Hoffmann et al. (2022), with diminishing returns at large scale:
$$R(x) \sim \mathcal{O}(N^{-\beta}), \quad \beta > 0. \tag{25}$$

- Since logarithmic growth dominates over power-law decay for large $N$, it follows that:
$$P(x) \gg R(x) \quad \text{for sufficiently large datasets.} \tag{26}$$

- The synergy term $S(P, R, x)$ is secondary and does not fundamentally shift this balance.

Thus, RL alone is unlikely to induce reasoning emergence; it only refines pretrained knowledge.   $\square$

### A.5 Summary of Theoretical Results

| Theorem | Key Finding |
|---|---|
| Sample Complexity (A.1) | RL requires quadratically more samples than supervised learning. |
| Reward Sparsity (A.2) | Meaningful rewards become exponentially rare as sequence length increases. |
| Compute Complexity (A.3) | RL scales quadratically in model size, making it computationally inefficient. |
| Pretraining vs. RL (A.4) | Pretraining dominates reasoning, while RL acts as fine-tuning. |

Table 4: Summary of theoretical findings.

