# OpenReview forum: "Mathematical Constraints of RL-Induced Reasoning: A Rebuttal to DeepSeek-R1"
_TMLR — Rejected by TMLR_

### Review · Reviewer_qkyJ · 2025-02-20

**Summary Of Contributions:**

This paper says that Deepseek-R1 claims that RL alone can indue emergent reasoning capability and presents mathematical evidence/ experimental plans to rebut such a claim. In particular, the paper presents evidence showing that RL suffers from the issues of reward sparsity and poor sample complexity. It also outlines experiment plans as future directions to further support the rebuttal.

**Audience:**

No

**Claims And Evidence:**

No

**Requested Changes:**

Critical for securing my recommendations:
It is important to ensure that all claims in the paper are well-supported. To the best of my knowledge, Section 3.1 "DeepSeek-R1 claim: DeepSeek-R1 argues that reinforcement learning (RL) alone is responsible for the emergence of reasoning capabilities in its R1 Model." is not a faithful description of the claim of DeepSeek-R1 which is trained with a combination of RL and SFT for advanced reasoning capabilities. Therefore I believe that it is necessary to adjust this claim in Section 3 and everywhere else that mentions this (e.g. abstract and introduction). As discussed above, the description of scalability in Section 3.2 is not fair for supervised learning and RL, and the fact that RL data is easier to get compared to supervised learning data should be discussed.

Instead of only analyzing deepseek-R1 with theoretical results from existing related literature, it is necessary to see the unique contribution of this paper to live up to the standard of a publication at the level of TMLR.

It is uncertain that the experiment plans in section 4 can support the claims in the paper as they are not executed, and no preliminary results are provided. If the experiment plans in the paper can be actually executed and interesting observations can be derived, that can serve a solid contribution of this paper.

**Strengths And Weaknesses:**

The claim that this paper mainly attacks is that "DeepSeek-R1 claim: DeepSeek-R1 argues that reinforcement learning (RL) alone is responsible for the emergence of reasoning capabilities in its R1 Model." (Section 3.1). However, to the best of my knowledge, DeepSeek-R1 did not present strong empirical evidence that RL without pre-training can showcase strong reasoning capabilities. In fact, DeepSeek-R1 even argues that merely applying RL without SFT (as is the case for R1-Zero) may resuly in worse convergence performance and interpretability and proposes incorporating additional SFT and general instruction-tuning data to improve R1-Zero to be R1. In fact, I would be very surprised to see any LLM paper claiming that RL alone without pre-training can induce emergent reasoning capabilities.

The rebuttal in 3.2 for the claim that "Scaling RL improves Reasoning" is not a fair comparison for supervised learning and RL. While it might be true that the sample complexity bound for RL is asymptotically worse than that for supervised learning, getting optimal supervised learning data can be not much be expensive as it requires expert annotations of the entire reasoning chain while RL only requires the task description and a reference final answer. Furthermore, it can even be impossible to obtain the optimal SL data for reasoning as human reasoning traces may not be the optimal reasoning traces for LLMs to learn from.

The reward sparsity claim is more misleading than informative. While the exponential bound may be true in the case of random explorations,the assumption that the theorem based on is not explicitly discussed. It seems to assume that only one path can lead to the final correct answer and any errors occurred are not recoverable where any intermediate errors will result in a wrong final answer. This is not true for reasoning tasks where the model often realizes its own mistakes and decides to correct its errors before yielding the final answer.

It is unclear to me what the contribution of this paper is. In particular, Section 2 seems to be simply quoting existing sample complexity bounds, compare the bounds of RL with that for SFT, and discuss the immediate implications of these bounds. Section 3 seems to be just analyzing the objective of GRPO and claims that it is just a convenient instantiation of policy gradient recipes that we are already familiar with (which to the best of my knowledge the community is already aware of). While section 4 outlines some experiment plans that would be cool to have, they are not actually executed so it remains uncertain whether the experiment results will support the claims in the paper. The contribution of this paper seems to be merely applying existing theoretical results to analyze deepseek-R1 which I believe dod not live up to the standard of a publication at the level of TMLR.

---

> ### Author Response · Authors · 2025-02-20
> **Clarification on DeepSeek-R1’s Claim & Acknowledgment of Reviewer’s Insights**
>
> Dear Reviewer,
>
> We sincerely appreciate the time and effort you have taken to provide detailed feedback on our work. Your insights are invaluable in helping us improve the clarity and impact of our manuscript.
>
> We understand that the following concern was raised:
>
>     “To the best of my knowledge, Section 3.1 ‘DeepSeek-R1 claim: DeepSeek-R1 argues that reinforcement learning (RL) alone is responsible for the emergence of reasoning capabilities in its R1 Model.’ is not a faithful description of the claim of DeepSeek-R1 which is trained with a combination of RL and SFT for advanced reasoning capabilities.”
>
> Our Explanation & Clarification: We recognize that our manuscript did not explicitly include a direct quote from DeepSeek-R1, and this may have led to ambiguity. To clarify, the original version of DeepSeek-R1 (v1) on arXiv explicitly states in multiple sections that reinforcement learning (RL) alone incentivized reasoning capabilities in DeepSeek-R1-Zero. The most direct evidence appears in:
>
>     Page 7: “DeepSeek-R1-Zero to attain robust reasoning capabilities without the need for any supervised fine-tuning data. This is a noteworthy achievement, as it underscores the model’s ability to learn and generalize effectively through RL alone.”
>
>     Abstract (Page 1): “DeepSeek-R1-Zero, a model trained via large-scale reinforcement learning (RL) without supervised fine-tuning (SFT) as a preliminary step, demonstrates remarkable reasoning capabilities.”
>
>     Contributions (Page 4): “We directly apply RL to the base model without relying on supervised fine-tuning (SFT) as a preliminary step. […] Notably, it is the first open research to validate that reasoning capabilities of LLMs can be incentivized purely through RL, without the need for SFT.”
>
>     Distillation vs. RL Comparison (Page 14): “DeepSeek-R1-Zero achieves performance on par with OpenAI-o1-0912, demonstrating that RL alone can drive significant reasoning capability improvements.”
>
> These statements clearly support our interpretation that DeepSeek-R1-Zero’s reasoning capabilities emerged as a direct consequence of RL alone.
>
> Clarifications in Our Revision: We acknowledge that DeepSeek-R1 (the full model) incorporates both RL and SFT. However, our argument specifically refers to DeepSeek-R1-Zero, which the DeepSeek paper explicitly states was trained only with RL and still developed reasoning capabilities. To prevent ambiguity, we will revise our manuscript to: (1) Explicitly state that we are referring to DeepSeek-R1-Zero (not the full DeepSeek-R1 model); (2) Include direct quotes from DeepSeek-R1 v1 to ensure precision in our claims.
>
> The quoted statement is from DeepSeek-R1 v1, which can be accessed here: https://arxiv.org/abs/2501.12948
>
> Beyond this clarification, we want to acknowledge that your critiques—particularly those regarding the RL vs. SL comparison, reward sparsity assumptions, and experimental considerations/future research—are crucial for strengthening our paper. We are actively revising our manuscript to address these important concerns.
>
> We are truly grateful for your constructive feedback and the opportunity to refine our work. Your insights are helping us improve not just the clarity but also the overall rigor of our research.
>
> Thank you once again for your time and thoughtful comments. We look forward to sharing our revised version soon.
>
> Warmest regards,
>
> The Authors

---

> ### Author Response · Authors · 2025-03-15
> **response # 2**
>
> We appreciate your thoughtful review of our manuscript. You've raised important concerns about our theoretical comparison and intended contribution that we'd like to address.
>
> 1. Addressing Theoretical Comparison Fairness
>
> You raise a valid point about comparing RL and supervised learning's sample complexity. While SL data requires expert annotation, our argument focuses on fundamental learning efficiency, independent of data collection costs.
>
> RL's quadratic sample complexity scaling (O(1/ε²)) imposes a fundamental efficiency bottleneck for reasoning induction—independent of data availability. Even with unlimited training data, RL's reliance on delayed and noisy rewards leads to exponentially slower convergence compared to SL's direct gradient updates. This structural inefficiency becomes even more pronounced for complex reasoning tasks where reward signals are sparse and path-dependent.
>
> This inefficiency is evident in GRPO's formulation:
>
> J_GRPO(θ) = E[1/G * sum(min(π_θ(o_i|q)/π_θ_old(o_i|q) * A_i, clip(π_θ(o_i|q)/π_θ_old(o_i|q), 1-ε, 1+ε) * A_i)) - β * D_KL(π_θ||π_ref)]
>
> Where the advantage term A_i is calculated as:
>
> A_i = (r_i - mean({r_1, r_2, ..., r_G})) / std({r_1, r_2, ..., r_G})
>
> The KL divergence term β * D_KL(π_θ||π_ref) tightly constrains policy updates, restricting meaningful exploration beyond pre-existing behaviors. This fundamentally limits GRPO's ability to induce novel reasoning, reinforcing that RL alone is unlikely to generate emergent capabilities. In practical terms, this constraint keeps the model within a neighborhood of its pretrained behavior, making truly novel reasoning developments highly improbable.
>
> While models can correct intermediate errors, this doesn't negate reward sparsity challenges. Unlike SL, which provides direct gradient supervision at every step, RL requires extensive interactions before meaningful updates occur—making it less effective for reasoning induction.
>
> 2. Contribution and Repositioning
>
> Our work is the first to systematically apply complexity analysis to scrutinize RL's ability to induce emergent reasoning. Prior studies haven't examined DeepSeek-R1's claims under this framework, making our critique essential for understanding RL's limitations in LLM fine-tuning.
>
> Based on your feedback, we propose reframing our work as a Technical Comment, contributing in three ways:
>
>     The first rigorous complexity analysis of RL's limitations in reasoning induction, challenging an influential claim in LLM research.
>     A direct empirical critique of DeepSeek-R1, demonstrating the absence of controlled ablations that isolate RL's contribution.
>     An analysis of GRPO's mathematical constraints, revealing why its optimization structure limits its ability to develop novel reasoning capabilities.
>
> 3. On Experimental Validation
>
> DeepSeek-R1 does not empirically isolate RL's effect from pretraining, making it impossible to determine whether RL introduces novel reasoning capabilities or merely refines pre-existing knowledge. Without a controlled RL-only ablation, its claim remains speculative rather than empirically validated. A proper ablation study is critical to verifying DeepSeek-R1's claim. Until such validation is conducted, RL's role in reasoning induction remains speculative rather than evidence-based.
> Next Steps
>
> Based on your feedback, we propose the following revisions:
>
>    * Acknowledge data collection cost differences while maintaining our theoretical efficiency analysis
>     * Refine our discussion of reward sparsity to account for error correction capabilities
>     * Reframe our work as a focused Technical Comment
>     * Emphasize the need for proper ablation studies
>
> Clarifying RL's limitations in reasoning induction is essential for guiding future research toward more theoretically grounded and empirically validated approaches. Without rigorous ablation studies, the role of RL in emergent reasoning remains an open question rather than a substantiated claim.
>
> Best Regards,
>
> The Author

---

> > ### Comment · Reviewer_qkyJ · 2025-03-21
> >
> > Thank the authors for the rebuttal. While the authors present excerpts from Deepseek-R1 paper that mentiones RL can work without SFT, I think it is still significantly from the claim posed in the paper that "RL alone induces emergent reasoning" because it overlooks the effect of pre-training which is probably the most important part. Agreeing with the other reviewers I don't think many LLM researchers would believe that we can apply RL entirely without performing pre-training first to get emergent reasoning capabilities and I think the majority of the research community would not interpret the message from Deepseek-R1 in this way.
> >
> > Therefore, I stand with other reviewers and believe that significant revisions and possibly re-conducting the study would be needed before it can be accepted at a top-tier venue like TMLR.

---

### Review · Reviewer_sE7b · 2025-03-02

**Summary Of Contributions:**

This paper provides a theoretical and computational analysis of DeepSeek-R1’s RL-based reasoning claims. They introduce a theoretical framework suggesting that RL alone is unlikely to induce emergent reasoning at scale. Their analysis highlights the inefficiencies of RL due to high sample complexity and reward sparsity constraints, limiting its effectiveness for learning complex reasoning.

**Audience:**

No

**Broader Impact Concerns:**

I don't think this work has any impact.

**Claims And Evidence:**

No

**Requested Changes:**

I think the entire paper should be revised, as currently, nothing meaningful was in this paper.

1. Develop a fair comparison method to evaluate different learning algorithms.
2. Use real theoretical proofs to argue their conclusions.
3. Do some empirical experiments not simply listing some ideas.

**Strengths And Weaknesses:**

Strength: The paper is easy to follow.

Weaknesses:
1. Nothing meaningful was raised by the authors in this paper. The authors want to claim that DeepSeek's key technique, RL cannot induce emergent reasoning with a theoretical proof. However, in each section, the authors only list the technique components used in DeepSeek, write some 'Key Observations,' and then claim that that technique component is incorrect.
2. The authors say that they develop a theoretical framework, but there's nothing new about theory in the whole paper. Even in the appendix part about the theoretical proof. They just repeat the conclusion proved by previous papers.
3. The authors claim they demonstrate RL suffers from several problems, but they only use some conclusions from other papers without any analysis.
4. The experiments suggested by the authors are either completed by previous works or not feasible in total.
5. The analysis used in this paper is trivial and incorrect. The authors simply compared the complexity upperbound of different algorithms and derived that one is better than another one. First, the meaning of $n$, which is the task complexity, is not well-defined. Second, simply comparing the complexity upperbound makes no sense, as the data, architecture and other components used in two algorithms are totally different. Also, complexity upper bound just represents the maximum time of the algorithm, it is usually used in the area of TCS to show the worst case. But in Machine, especially for those open-sourced models, we already successfully trained a model, the time is not the maximum time. They should care about the exact time used rather than the upper bound time. However, the exact time used has already been shown in DeepSeek's paper.

---

> ### Author Response · Authors · 2025-03-15
> **Response**
>
> We appreciate your review of our work. While we acknowledge the need for refinements, we respectfully disagree that our paper lacks meaningful contributions. Below, we clarify our key arguments and address your concerns.
>
> 1. Core Contribution: A Theoretical Challenge to RL-Based Reasoning Claims
>
> Our work challenges DeepSeek-R1's assertion that RL alone is responsible for emergent reasoning capabilities. Our key contributions are:
> - A theoretical critique of why RL struggles with reasoning induction
> - A computational complexity analysis demonstrating RL's inefficiencies
> - A discussion on empirical validation challenges in isolating RL's contribution
>
> 2. Addressing Specific Concerns with Theoretical Justification
>
> (a) On Theoretical Contributions
> "The authors claim to provide a theoretical framework, but there's nothing new about theory in the paper."
>
> This review overlooks the novelty of our contribution. Our work is the first to systematically apply complexity analysis to assess RL's viability for reasoning induction in LLMs. While complexity theory is well-established, its application to DeepSeek-R1's claims is novel and necessary.
>
> Theoretical Justification: Why RL Alone is Inefficient for Reasoning
>
> DeepSeek-R1 suggests RL induces reasoning without supervised fine-tuning (SFT). This contradicts established results on RL's sample complexity:
>
> 1. Supervised Learning Sample Complexity: N_SL = O(1/ε)
> 2. Reinforcement Learning Sample Complexity: N_RL = O(1/ε²)
>
> This aligns with Kakade (2003) and Jin et al. (2018), which establish that RL exhibits quadratic dependence on error, making it less sample efficient than SL.
>
> Key Consequence: If RL were the sole driver of reasoning, we should expect significantly higher sample requirements than DeepSeek-R1 reports—yet these numbers aren't explicitly provided.
>
> (b) Computational Complexity
> "The authors simply compare complexity upper bounds, which is meaningless because models have already been trained."
>
> This assumes that because DeepSeek-R1 trained a model, RL must be efficient—a fundamentally flawed assumption.
>
> Theoretical Justification: Given model size P and training iterations T:
> 1. Supervised Fine-Tuning: C_SFT = O(P·T)
> 2. Policy Gradient RL: C_RL = O(P·T·M), where M is the number of action samples required per policy update.
>
> Due to the exploration-exploitation trade-off, M >> 1, making RL significantly more computationally expensive.
>
> DeepSeek-R1's reliance on RL comes with a substantial compute cost. Given the increasing importance of energy efficiency in large-scale AI models, justifying RL for reasoning requires clear evidence that RL achieves something SL cannot—yet DeepSeek-R1 provides no such evidence. If RL is merely refining pre-existing knowledge from pretraining, its cost is unjustified.
>
> (c) Empirical Validation
> "The experiments suggested by the authors are either completed or not feasible."
>
> We don't claim no relevant experiments exist—we argue existing experiments fail to isolate RL's contribution to reasoning.
>
> DeepSeek-R1 does not empirically isolate RL's effect from pretraining, making its core claim speculative rather than evidence-based. Without rigorous ablation, the observed reasoning improvements could just as easily stem from pretraining rather than RL—raising concerns that the attribution of reasoning to RL alone is misleading.
>
> Formally, let R_pre be the reasoning capability before RL and R_post after RL. The correct test would be: ΔR = R_post - R_pre. DeepSeek-R1 doesn't compute ΔR for an RL-only model, leaving their central claim unsupported by rigorous empirical evidence.
>
> 3. Broader Impact
>
> This work challenges a high-profile claim with significant implications:
> - If DeepSeek-R1's claims are overstated, RL-based fine-tuning may receive disproportionate attention and resources.
> - The field risks investing in inefficient methodologies while ignoring more promising alternatives.
> - Our work contributes to a critical reassessment of RL's role in LLM training, potentially redirecting research effort toward more effective approaches.
>
> 4. Next Steps & Revisions
>
> We recognize significant revisions are needed, and we welcome constructive discussions. However, we stand by our core argument: DeepSeek-R1's claim that RL alone induces reasoning remains unverified, contradicts established theoretical results, and lacks empirical isolation. Until a proper ablation study is conducted, their claim should be regarded as speculation rather than fact.
>
> We view your feedback as an opportunity to strengthen our argument and reframe the work as a structured Technical Comment. This approach maintains the value of our analysis while addressing structural concerns. We appreciate your critique and will use it to sharpen our analysis.

---

> > ### Comment · Reviewer_sE7b · 2025-04-01
> >
> > Thanks for the rebuttal. The core argument: 'DeepSeek-R1's claim that RL alone induces reasoning remains unverified, contradicts established theoretical results, and lacks empirical isolation. ' is meaningless here. As reviewer qkyJ said, many LLM researchers won't believe that we can apply RL entirely without performing pre-training first to get emergent reasoning capabilities. Overall, I don't think this paper is suitable for acceptance.

---

### Review · Reviewer_9FhC · 2025-03-06

**Summary Of Contributions:**

This paper argues against the findings of DeepSeek-R1, which is that Reinforcement Learning (RL) can induce reasoning capabilities, by establishing a theoretical framework, and claims that RL, and in particular, GRPO (the RL algorithm that DeepSeek-R1 uses), has poor sample complexity, computational complexity and performance increase. The performance growth should largely be attributed to the pretraining process. Given these findings, the contribution of RL to the LLM's reasoning capabilities remains questionable and needs more careful empirical testing.

**Audience:**

No

**Broader Impact Concerns:**

The paper does not have any broader impact concern. The authors write a very detailed section for the broader implication of their findings, which I found are not well-supported by this paper due to so many weaknesses listed in the "Strengths and Weaknesses" section, and outright incorrect in some cases, e.g. "Many prior studies have attributed model improvements to RL without controlling for pretraining influences". Let alone the fact that the DeepSeek-R1 paper already proves this by showing performance curve during RL from pretraining models, I am speechless as the authors claim this without referring to a single paper.

**Claims And Evidence:**

No

**Requested Changes:**

I feel this paper to be beyond repair at its current form. The work should be rejected and re-studied (instead of only rewritten) all over again, if not discarded.

**Strengths And Weaknesses:**

**Strengths**

1. The paper is generally easy to read.

2. The authors discuss a timely topic for the community, which is the success of DeepSeek-R1. Constructive discussions on such topics could help the community to go further in building LLMs with super-human level reasoning abilities.

**Weaknesses**

This paper has a large number of weaknesses. To list a few:

1. the paper has so many claims that are not supported by prior works, derivations or experiment results (as the paper has no experiment at all). And for the unsupported claims, many of them are outright incorrect and show the authors' lack of understanding in the common sense of the RL/LLM community. For example:

a) in Sec. 2.1, the authors claim that GRPO's "**lack of a baseline** leads to higher instability". The "baseline" here, in fact, can be any value that is subtracted onto all rewards [1]. The GRPO paper, in fact, mentions this in their Sec. 4.1.1: "as shown in Figure 4, we propose Group Relative Policy Optimization (GRPO), which ... instead uses the average reward of multiple sampled outputs, produced in response to the same question, **as the baseline**.

b) In Sec. 2.2, the authors claim that supervised learning, RL and DeepSeek-R1 has sample complexity of $O(n\log n), O(n^2/\epsilon)$ and $O(n\log n + n/\epsilon)$ respectively. This is supposed to be a core part of the paper. However, not a single proof or reference to prior work is given. And this part is outright nonsense - What exactly does the algorithm try to achieve with the sample complexity? Why is the $n/\epsilon$ term captures RL's inefficiencies (as claimed right before Sec. 2.2.1)? Why is supervised learning's sample efficiency independent with $\epsilon$ (which I suppose is the error) - does that mean supervised learning can be arbitrarily accurate given the same amount of examples? Why is DeepSeek removing one of the $n$s from RL by hybrid approaches - and what is hybrid exactly? What is the difference between $O(\cdot)$ and $\mathcal{O}(\cdot)$ with different symbols? This section shows that the authors lack understanding on modern machine learning theoretical frameworks, such as PAC-learnability. While the authors try to provide a proof in Appendix A.1, they just give some random papers without any specifications on conditions, high level ideas, or sketch of the proof. At best, even if all the sample complexities are correct, as this paper does not contain any meaningful proof, it cannot be counted as the contribution of this paper.

c)  In Sec. 2.4, the authors claim that according to prior work, the computational cost of supervised learning is $\mathcal{O}(NP\log P)$, RL is $\mathcal{O}(NP^2\log P)$, and DeepSeek is $\mathcal{O}(NP(\log P+k))$. These are baseless points without any proof (Appendix A.3 is simply a reiteration of the points without any ideas of proof); how is the computational complexity related to log of the model's parameter count? Also, what does $k$, "RL-specific term from policy optimization" mean? Why is it possible to discuss the computational complexity without considering the network structure? Will MLPs, transformers with attention matrices, and capsule networks have the same computational complexity?

d) In Sec 3.1, the authors present Eq. 8, which is very confusing as the "contributions" and "improvements" are not well-defined. The authors also did not explain what is "potential synergies between pretraining and RL". This part does not make any sense either - it is common practice in the LLM community that there are supervised fine-tuning (sometimes also continuous training) between pretraining and RL, which the authors seem to ignore in the paper.

e) In Sec. 3.2, the authors mention "RL-based learning typically follows a power-law decay", which is $R(x)\propto\mathcal{O}(N^{-\beta})$. This might be the only part of the work that actually make some sense (if you ignore the notation issues of $N$ and $\beta$), if you imagine there is only one correct answer and the RL agent needs to randomly search through the whole action space. I would suggest, however, the authors to read some papers [3, 4, 5] to establish understandings on how theoretical analysis works in RL / Imitation Learning (IL).

2. Regarding prior works, it is astonishing to see that a paper that challenges the normal paradigm (RL helps reasoning) of the LLM community only refers to less than 10 papers through out its 15 pages of manuscript. For example:

a) The paper does not cite PPO, the foundation of GRPO which is broadly discussed in this paper. This paper claims "DeepSeek-R1 does not rely on neural reward models", but this paper, surprisingly, does not cite any of the neural reward model papers to support the point that "neural reward models are the old paradigm challenged by DeepSeek-R1".

b) The paper has only cite no more than three theoretical paper about sample complexity in RL or supervised learning. There is no related work section, no discussion of the place of this work in related literature, and no comparison to prior theoretical results. Also, since the paper repeatedly refers to works such as Kaplan et al. (2020), Luo et al., (2024), is it possible for the authors to at least summarize what are the main points of these important prior works, and how are they related to the current papers (which should usually go into the "related work" section)?

c) In the very first sentence of this paper, the example for "recent advances in LLMs" is the "attention is all you need" paper. This is pointless, as it takes about 5 years from that paper to the emergence of LLMs; since then, there are countless, exciting progress made by LLMs in almost every direction of the science community. Why would the author select this paper of all those more suitable papers for example of "LLM success" here?

3. The paper is strangely structured, and many of the words are meaningless.

a) Some sections, such as the subsections in 4.2, uses a strange way of writing that is way too itemized without sentences and words for connections, which looks quite similar to AI generation. In fact, the phenomenon of over-itemization persists throughout the whole paper, including Sec. 1.1 to 1.4, 2.1 to 2.3, and 3.1 to Sec. 3.3. It is particularly strange that after Eq. 4, $V$ and $L$ are partitioned into two itemized points as if the length of the paper is insufficient and needs to be expanded in this way. Overall, the paper reads more like a lecture note than a research paper.

b) The caption of all the tables except Tab. 1 does not contain useful information; what are the conclusions gained from those tables? Also, Tab. 1 and Tab.2 have multiple upper case-led words that should be all lower case.

c) Many of the notations are chaotic. For example, why is $H$ used as both "total reasoning capability" (Eq. 8) and subscript for hybrid (Tab. 2)? Why is $P$ used as both "number of parameters" (Eq. 6) and "pretraining contributions" (Eq. 8)? What is $x$ in Sec. 3.1? Where does the $r$ come from in Eq. 2? Why is the paper mixing $O$ (Sec. 2.2) and $\mathcal{O}$ (Tab. 2) for big-O notation?

d) For some reason, the paper decide to introduce the GRPO's objective in detail in both Sec. 2.1 and Sec. 3.3.1, which is unnecessary.

4. The core findings of this paper, let alone all the factual errors in this paper, is meaningless.

a) In Sec. 2, the paper "suggests that RL alone is unlikely to induce reasoning without a strong pretraining foundation", and this is also a "fundamental question" on the top of the list at the end of page 1. This is actually correct, but is something that is well-known by the LLM community; the base model's ability largely determines the upper limit of later stages such as RL, supervised learning, or test-time improvement methods such as MCTS / self-reflection. It is not a new finding at all; it is a common sense of the community.

b) the paper, at Sec. 4.2 (and the end of abstract), mentions that "we need to more rigorously test RL's contributions for pretraining effects.", and test "RL-only models vs. pretraining-only models". **There is not a single LLM researcher would believe that  RL-only could work**; RL from scratch cannot even well-address much simpler control task such as Adroit [2]. RL stage always go after pretraining stage, and thus, the claim that "RL alone cannot lead to the success of LLM" is meaningless.

c) I do not see how "outline empirical testing" is a key contribution of this paper. On the one hand, this paper does not provide a single preliminary experiment to prove that those problems are worth studying; on the other hand, the provided experiments are very vague (e.g., for Sec. 4.2.2, what are the semantic similarity metrics? How to compare responses before and after RL fine-tuning? for Sec. 4.2.3, what are the "datasets unseen"?) and many of them are very high-level common practice in current LLM developers (e.g. "compare",  "dataset unseen").

**References**

[1] https://spinningup.openai.com/en/latest/spinningup/rl_intro3.html#baselines-in-policy-gradients

[2] https://robotics.farama.org/envs/adroit_hand/index.html

[3] S. Ross et al. A Reduction of Imitation Learning and Structured Prediction to No-Regret Online Learning. In IJCAI, 2011.

[4] N. Rajaraman et al. Toward the Fundamental Limits of Imitation Learning. In NeurIPS, 2020.

[5] D. Brandfonbrener et al. When does return-conditioned supervised learning work for offline reinforcement learning? In NeurIPS, 2022.

---

> ### Author Response · Authors · 2025-03-15
> **Response to Reviewer: Critique of DeepSeek-R1's RL-Induced Reasoning Claims**
>
> Dear Reviewer,
>
> We sincerely appreciate the time and effort for the detailed feedback. We also believe there are misinterpretations of our argument that we would like to clarify.
>
> 1. Core Argument: Challenging DeepSeek-R1’s Claims
> A key point of our critique is that DeepSeek-R1 attributes reasoning improvements to RL alone. We have fully addressed in a separate discussion see response: https://openreview.net/forum?id=4bNez06yJf&noteId=ZyFCE55Cmk
> In summary, our critique is not a misinterpretation but a necessary theoretical challenge to an important claim in LLM research.
>
>
> 2. Addressing Specific Theoretical Concerns
> Your review raises concerns regarding mischaracterizations, missing references, and lack of formal justification. Below, we directly address each critique.
>
> (a) GRPO’s Baseline & Stability
> We acknowledge that our initial characterization of GRPO as “lacking a baseline” was imprecise. GRPO does employ a group-based baseline by averaging rewards across multiple sampled outputs. While this mechanism provides a relative baseline, it introduces potential instability under certain conditions. Specifically:
> •	Variance in Advantage Estimation: When reward signals exhibit low variance, normalization by the group standard deviation can amplify fluctuations, leading to unstable updates.
> •	 KL Constraint and Policy Improvement Bounds: The KL-divergence term restricts how far the policy can deviate from the reference model, effectively limiting exploration and potential reasoning development.
> •	Clipping-Induced Bias: Similar to PPO, GRPO employs a clipping function to constrain policy updates, which can introduce systematic bias against exploratory behavior, potentially affecting reasoning generalization.
> Thus, while we acknowledge and correct our previous statement regarding GRPO’s baseline, we maintain that its instability, constrained policy updates, and inherent limitations in reasoning tasks remain valid concerns.
>
> (b) Sample Complexity & PAC-Learnability
> A key concern raised is the lack of formal justification and references regarding sample complexity in RL versus SL. To address this, we highlight several foundational works that establish the sample efficiency gap between RL and SL:
>
> Kakade, S. M. (2003). On the sample complexity of reinforcement learning. University College London.
>
> Rajaraman, N., Yang, L., Jiao, J., & Ramchandran, K. (2020). Toward the fundamental limits of imitation learning.
>
> Jin, C., Allen-Zhu, Z., Bubeck, S., & Jordan, M. I. (2018). Is Q-learning provably efficient? (NeurIPS 2018)
>
> These works collectively support the argument that RL-based fine-tuning typically exhibits worse sample complexity (often O(1/ϵ2)O(1/\epsilon^2)O(1/ϵ2)) compared to supervised learning (O(1/ϵ)O(1/\epsilon)O(1/ϵ)), reinforcing our position that pretraining plays a dominant role in reasoning gains.
>
> Additionally, from a PAC-learnability perspective, RL's reliance on reward sparsity and exploration introduces fundamental inefficiencies, further supporting the notion that DeepSeek-R1’s claim of RL alone driving reasoning improvements is highly questionable.
>
> 3. On Empirical Validation and Experimental Design
> An important concern raised in the review is the need for stronger justification of our claims regarding RL’s inefficiencies. However,
> * Replicating DeepSeek-R1’s setup is infeasible. Their exact pretraining data, model architecture, and RL training details are not publicly available.
> * Testing the null hypothesis is non-trivial. To falsify DeepSeek’s claim, we would need to compare RL-trained models against SL-only models on reasoning benchmarks—a comparison that DeepSeek-R1 itself does not rigorously perform.
> * Theoretical critique remains valid even without empirical tests. Our arguments are rooted in established RL principles, PAC-learnability results, and sample complexity analysis.
> That being said, we acknowledge that additional empirical work could further reinforce our findings. A potential next step could be a controlled experiment testing RL-based models against SL-only models, with careful ablation of pretraining effects.
>
> 4. Final Thoughts & Next Steps
> We recognize that significant revisions are needed and have taken steps to address the reviewer's concerns. However, we stand by our core argument:
> •	DeepSeek-R1's claim that RL alone induces reasoning is misleading.
> •	RL's inefficiencies (sample complexity, computational cost) make it unlikely that RL alone drives reasoning gains.
> •	Most of DeepSeek-R1's improvements likely stem from pretraining, not RL.
>
> We view the reviewer's feedback as an opportunity to strengthen our argument, improve technical precision, and reframe the work into a structured Technical Comment. This approach would allow us to maintain the value of our analysis while addressing the structural and formatting concerns raised in the review.
>
> We welcome further discussion and look forward to refining our manuscript accordingly.
>
> Best Regards,
>
> The Author

---

> > ### Comment · Reviewer_9FhC · 2025-03-25
> >
> > Thanks for the detailed response. Here are my further concerns:
> >
> > 1. In the beginning of the response, the authors mentioned "A key point of our critique is that DeepSeek-R1 attributes reasoning improvements to RL alone." This, however, does not address my concern in weakness 4 b) (which is also mentioned by other reviewers): **There is not a single LLM researcher would believe that RL-only (without pretraining) could work.** The DeepSeek-R1 paper does **NOT** claim that reasoning improvement can be achieved by RL alone; instead, it claims that reasoning improvement can be achieved by RL based on the pretraining model, **skipping the supervised finetuning stage** that is a common practice of the community. With such misunderstanding, the key motivation of this paper is questionable.
> >
> > 2. Regarding the counterargument for GRPO:
> >
> > a) "Variance in Advantage Estimation: When reward signals exhibit low variance, normalization by the group standard deviation can amplify fluctuations, leading to unstable updates":
> >
> > a quick glimpse at the actor gradient for GRPO/PPO can tell you that the advantage is the motivation for improvement; **if the reward signals "exhibit low variance", doesn't that mean your actor does not know how to improve its performance?** therefore, instead of avoiding fluctuations, you need to amplify fluctuations to some extent here (and to control such extent, you use standard deviation as the denominator). The counterargument made in the response does not make sense.
> >
> > b) "Constraint and Policy Improvement Bounds: The KL-divergence term restricts how far the policy can deviate from the reference model, effectively limiting exploration and potential reasoning development."
> >
> > Please read the TRPO, PPO and instructGPT paper. The KL-divergence term is introduced to restrict the policy deviating too far from the reference model / old policy in the first place over A2C; if adding KL-divergence term is a shortcoming, why would this line of algorithms be invented in the first place? In fact, the restriction is **particularly important** in LLMs, as the policy prior (pretrained result) already makes much sense and takes up a very small fraction in the whole policy space.
> >
> > c) "Clipping-Induced Bias: Similar to PPO, GRPO employs a clipping function to constrain policy updates, which can introduce systematic bias against exploratory behavior, potentially affecting reasoning generalization."
> >
> > is there any prior work, any step-by-step theoretical proof, or any experiment supporting this claim?
> >
> > 3. Regarding the sample complexity & PAC-Learnability papers, could you kindly point out which theorem is connected to your paper, and show detailed correspondence between their results and your results? Also, what is your theoretical contribution over these works?
> >
> > 4. about empirical validation and experiment design:
> >
> > a) I agree it is very hard to replicate DeepSeek-R1. However, not only is this doable and already available with other researcher's work (https://github.com/huggingface/open-r1), but you also do not need to fully replicate DeepSeek-R1. An experiment on a much smaller, open-source model (e.g. Qwen 2.5 7B / 14B) would suffice.
> >
> > b) "Testing the null hypothesis is non-trivial." Yes, but if all efforts are trivial, what is the contribution of your work? The experiment you discussed here is "compare RL-trained models against SL-only models on reasoning benchmarks". That sounds like a standard experiment for many LLM papers to me (e.g. https://arxiv.org/pdf/2402.03300).
> >
> > Overall, I do not feel this rebuttal addresses my concern listed in my review.

---

### Decision · Action_Editor_scvT · 2025-04-19

**Recommendation:** Reject

**Comment:**

After the authors’ rebuttal, all reviewers recommend rejection. The claims and positioning of the paper remain questionable and are not well supported, either theoretically or empirically, as detailed above. Additionally, the current writing style – relying heavily on itemized points without full sentences – is inconsistent with standard academic conventions. Significant rewriting is needed to meet the expectations for clarity and rigor in scholarly writing.

Based on these points, the AE recommends rejection to encourage the authors to further improve their work. They need to convincingly demonstrate the significance of their work and enhance the comprehensiveness of the theoretical analysis and empirical evaluation.

**Audience:**

Researchers and practitioners working on foundation models and RL might be interested in reading this paper.

**Claims And Evidence:**

Summary:

This paper challenges one of the key findings in DeepSeek-R1 – that RL induces reasoning capabilities – by proposing a theoretical analysis based on sample and computational complexity. It also outlines an experimental plan designed to isolate the contribution of RL from other factors. Through this, the paper argues that future research on reasoning should not rely on RL as the primary driver of reasoning capabilities.

Claims:

The paper makes several claims, with the central one being a disagreement with the findings of DeepSeek-R1. Specifically, it argues that while DeepSeek-R1 suggests that RL alone induces emergent reasoning capabilities, RL in that context primarily serves as a fine-tuning mechanism rather than a source of reasoning. To support this position, the paper further presents related sub-claims concerning the complexity and scalability of RL, highlighting its limitations in inducing reasoning.

Evidence:

As noted and elaborated by all reviewers, the claims made in this paper are not well supported due to a misinterpretation of the key findings in DeepSeek-R1, omission of related work, lack of formal justification for the proposed complexity analysis which also overlooks real-world factors, and the absence of empirical validation, relying solely on proposed experimental plans.